# Deciphering the Alphabet of Disorder—Glu and Asp Act Differently on Local but Not Global Properties

**DOI:** 10.3390/biom12101426

**Published:** 2022-10-04

**Authors:** Mette Ahrensback Roesgaard, Jeppe E. Lundsgaard, Estella A. Newcombe, Nina L. Jacobsen, Francesco Pesce, Emil E. Tranchant, Søren Lindemose, Andreas Prestel, Rasmus Hartmann-Petersen, Kresten Lindorff-Larsen, Birthe B. Kragelund

**Affiliations:** Linderstrøm-Lang Centre for Protein Science, Department of Biology, University of Copenhagen, Ole Maaløes Vej 5, 2200 Copenhagen, Denmark

**Keywords:** Dss1, intrinsically disordered protein, IDPs, molecular dynamics, NMR, sequence composition, SAXS

## Abstract

Compared to folded proteins, the sequences of intrinsically disordered proteins (IDPs) are enriched in polar and charged amino acids. Glutamate is one of the most enriched amino acids in IDPs, while the chemically similar amino acid aspartate is less enriched. So far, the underlying functional differences between glutamates and aspartates in IDPs remain poorly understood. In this study, we examine the differential effects of aspartate and glutamates in IDPs by comparing the function and conformational ensemble of glutamate and aspartate variants of the disordered protein Dss1, using a range of assays, including interaction studies, nuclear magnetic resonance spectroscopy, small-angle X-ray scattering and molecular dynamics simulation. First, we analyze the sequences of the rapidly growing database of experimentally verified IDPs (DisProt) and show that glutamate enrichment is not caused by a taxonomy bias in IDPs. From analyses of local and global structural properties as well as cell growth and protein-protein interactions using a model acidic IDP from yeast and three Glu/Asp variants, we find that while the Glu/Asp variants support similar function and global dimensions, the variants differ in their binding affinities and population of local transient structural elements. We speculate that these local structural differences may play roles in functional diversity, where glutamates can support increased helicity, important for folding and binding, while aspartates support extended structures and form helical caps, as well as playing more relevant roles in, e.g., transactivation domains and ion-binding.

## 1. Introduction

Intrinsically disordered proteins (IDPs) are involved in various cellular processes, including cell cycle regulation, cellular signaling and protein degradation. The malfunctioning or aggregation of IDPs can cause diseases such as cancer, diabetes, Parkinson’s disease and Alzheimer’s disease [1,2]. IDPs are characterized by not adapting one specific spatial structure, instead fluctuating between a large number of conformations, distinguishing them from folded proteins. While the sequence-structure-function paradigm is by now well established for folded proteins, the link between sequence and function for IDPs is still poorly understood [3]. Understanding the relationship between sequence, conformational ensemble and function is important for understanding the structural basis of fundamental processes in life and for reaching treatment options for complex diseases.

The pioneering bioinformatics work in the early 2000s recognized IDPs as a separate class of proteins and showed that the amino acid composition of IDPs was distinct from that of folded proteins. The most significant characteristics were a low content of hydrophobic residues and a high content of charged and polar residues [4,5]. Some amino acids were found to be significantly more enriched or depleted in IDPs [6], and subsequent studies by Uversky et al. from 2007 and 2013 [7,8] found similar enrichment. Glutamate is one of the most enriched amino acids in IDPs, while aspartate, which differs from glutamate only by having one less methylene group in the sidechain (Figure 1A) was surprisingly found not to be substantially enriched. The same can be observed for the two similar amino acids, glutamine and asparagine, where glutamine is more enriched in IDPs than asparagine, with no explanation for these differences having been provided. One possible explanation for the difference in the enrichment of glutamate and aspartate could be that the two amino acids have distinct disorder-promoting properties; however, the difference could also be related to differences in the behavior of aspartate and glutamate in the reference set of folded proteins. The extra methylene group in glutamate compared to aspartate results in a larger conformational space, which could favor solvent-exposed conformations over buried conformations. Glutamate is also more frequently observed in α helices than aspartate [9] and is more helix-stabilizing, because the carboxyl group is further from the backbone and thus imposes fewer restraints on the conformational space of the residues in the helix [10]. Transient helicity is frequently observed in IDPs, and a larger population of transient helicity in the free state has been linked to faster binding to a target [11,12], which could be a structural and functional explanation of the observed enrichment of glutamate. However, the analysis of the sequence composition of IDPs could be biased by factors that are not related to structure. Most of the proteins in the intrinsic disorder database DisProt [13,14,15] are eukaryotic, and more than a third of the proteins are human, while the Protein Data Bank (PDB) [16] contains many bacterial proteins, as of October 2022. A bias in sequence composition when comparing DisProt to the PDB could therefore be because the sequence composition from eukaryotes differs from bacteria or because the human protein sequence composition is distinct from the average sequence composition. Further, since the early studies by Uversky and co-workers [7,8], the number of experimentally verified disordered sequences deposited in DisProt has increased five times, as of July 2022.

In this work, we revisited the sequence enrichment profile of IDPs using a larger dataset of disordered proteins and investigated if compositional bias arising from other factors than structural properties exists, including species variability. The analyses confirmed the previous observation of a greater enrichment of glutamates than aspartates in IDPs, and we did not find significant differences across species. To address if glutamate is more disorder-promoting than aspartate, we designed Glu/Asp variants of a model IDP. Deleted in split hand/split foot 1 (Dss1) from *S. pombe* was chosen because it is a well-studied IDP with a high content of negatively charged residues, many of which are fully conserved, showing preferences for either Glu or Asp, as shown in Figure 1B. Dss1 is known to have multiple interaction partners [19] and is found in various complexes, including the 26S proteasome [20,21]. Dss1 adapts different conformations when bound in different complexes but retains disordered regions upon binding [22]. The C-terminal region adapts an α-helical conformation when bound to, e.g., the proteasome and the T-REX complex, and it also exists transiently when free in solution [23]. In the proteasome, Dss1 functions as a ubiquitin-receptor, binding polyubiquitylated substrates destined for degradation. Thus, while Dss1 binds mono-ubiquitin, it can also bind chains of ubiquitin, exploiting two disordered ubiquitin-binding motifs [24] (UBS) I from D38–D49 and UBS II from D16–N25, with UBS I has the strongest affinity for ubiquitin [23]. A transiently formed C-terminal helix in Dss1 interacts dynamically with UBS I, and this fold-back structure may function to shield the binding site and thereby regulate binding partner interaction [19]. From the alignments, the sequences of UBS I, UBS II and the C-terminal helix are highly conserved within the family (Figure 1B). By designing variants of Dss1 containing only aspartate, only glutamate or Glu/Asp swaps, we addressed the impact of Glu/Asp variants on the function of Dss1 both by growth assays in yeast and by measuring ubiquitin binding. Additionally, we investigated the conformational ensembles of the Dss1 Glu/Asp variants using nuclear magnetic resonance (NMR) spectroscopy, small-angle X-ray scattering (SAXS) and molecular dynamics (MD) simulation. We found that Glu/Asp variants do not impair yeast growth and that the proteins have similar global dimensions and bind ubiquitin, but the specific Glu/Asp pattern in UBS I influences the binding affinity. Additionally, the length and population of the transient C-terminal helix was found to depend on the presence of N-terminal helix-capping aspartates and the presence of a glutamate in the center of the helix. We concluded that glutamate and aspartate confer different functional and structural properties in an IDP acting on a local scale, which may contribute to the observed higher enrichment of glutamate in IDPs.

## 2. Materials and Methods

Composition profiles were constructed using the web tool developed by Uversky et al., Composition Profiler [8], using 50,000 boot strap iterations and a statistical significance value of 0.05. The query dataset consisted of all non-redundant, non-ambiguous and non-obsolete sequences from DisProt v. 9.0.1 [13,14,15], and for the background dataset, the reviewed part of the database UniProt-UniParc (release 2022_02) [25] was used to obtain a non-redundant set of protein sequences with natural origin from the Protein Data Bank (PDB) [16].

We designed three variants of *S. pombe* Dss1 consisting of a variant with all aspartates substituted for glutamate (All-E), all glutamates substituted for aspartate (All-D), all glutamates substituted for aspartate and all aspartates substituted for glutamate (Swap) and included the wildtype (WT) for comparison. All variants carried a substitution of asparagine for cysteine at the C-terminus, and codons were optimized for *E. coli* expression. Additionally, peptides corresponding to the transient helical region of Dss1 (residues 51–69) with substitutions at key positions were designed and purchased from Pepscan (The Netherlands).

### 2.1. Yeast Strains and Techniques

The *dss1*Δ strain has been described before [23]. The pDUAL vector [26] was used for the expression of *dss1*^+^ and the *dss1* variants carrying N-terminal HFG (6His, Flag, green fluorescent protein (GFP)) tags. Cloning and mutagenesis were performed using Genscript. The yeast strains were transformed using lithium acetate [27]. Growth assays were performed on Edinburgh minimal media (EMM2) (San Diego, CA, USA) as described previously [23]. The preparation of cell lysate samples for SDS-PAGE was performed using trichloroacetic acid and glass beads as described previously [23]. The samples were separated by SDS-PAGE on 12.5% acrylamide gels and transferred to 0.2 μm pore-size nitrocellulose membranes (Advantec, Tokyo, Japan). The antibody was anti-GFP (1:1000, Chromotek, Planegg, Germany, Cat# 3H9). Secondary horseradish-peroxidase-conjugated antibodies were from Dako Cytomation. Equal loading was checked using stain-free imaging with 0.5% trichloroethanol (Sigma, St. Louis, MO, USA).

### 2.2. Protein Purification

All four variants of *S. pombe* Dss1 were designed to encode an N-terminal His_6_-SUMO tag to be cleaved with ubiquitin-like protein protease 1 (ULP1) following initial purification with a nickel column [28]. All four variants were purified with isotope labeling, as described in previous work [29], resulting in lyophilized pure protein, ready for resuspension in the buffer of choice. His_6_-SUMO ubiquitin was purified for protein interaction studies with Dss1, similarly to what has been described in previous work [30].

### 2.3. NMR Assignment

Assignments of Dss1 variants were carried out from a series of ^1^H-^15^N HSQC, HNCACB, HN(CO)CACB, HNCO, HN(CA)CO and HN(CA)NNH spectra, as described previously for WT Dss1 [19]. Spectra were recorded either on a Bruker Avance III HD 750 MHz spectrometer, with a Bruker proton-optimized triple-resonance 5mm TCI cryoprobe, or on a Bruker Avance Neo 800 MHz spectrometer, with a Bruker 5 mm CPTXO cryoprobe. Spectra were processed in TopSpin version 3.6.2 (Bruker, Fällanden, Switzerland), transformed using qMDD version 3.2 [8] and processed through nmrDraw version 9.9 [31]. Analysis was carried out using CcpNmr Analysis version 2.5.0 [32]. Samples were transferred to single-use LabScape Essence 5 mm NMR Tubes (Bruker, Switzerland), and all spectra were recorded at 10 °C, unless noted otherwise.

### 2.4. Secondary Chemical Shift (SCS) Analysis

Random coil chemical shifts were computed for variants using a predictor based on previous work [33,34]. Conditions (10 °C and pH 7.4) were specified as input together with the sequence. From the observed assigned chemical shifts (*δ_observed_*) and the predicted random coil chemical shifts (*δ_random coil_*), the secondary chemical shifts (*SCS* or ∆*δ*) for different nuclei were calculated [35]:(1)$$SCS=∆\delta =\delta_{observed}-\delta_{random\;coil}$$

The resulting nuclei-specific datasets were used as an indication of local and structural properties in the different Dss1 variants.

### 2.5. Ubiquitin Binding

The Dss1 variants and ubiquitin were brought to the same buffer solution, with a final concentration of 20 mM Na_2_HPO_4_/NaH_2_PO_4_ and 150 mM NaCl at pH 7.4; 10% (*v*/*v*) D_2_O, 1% (*v*/*v*) sodium 2,2-dimethyl-2-silapentane-5-sulfonate (DSS) and 5 mM dithiothreitol (DTT) were added, and the pH was readjusted to 7.4. Ubiquitin and the Dss1 variant were mixed, resulting in 1:1, 1:3, 1:6, 1:9, 1:20 and 1:40 molar equivalents of ubiquitin, keeping the Dss1 variant concentration at 50 µM in all titration points. The ^1^H-1D and ^1^H-^15^N heteronuclear single quantum coherence (HSQC) spectra were recorded for all titration points as well as for a sample of 50 µM of a Dss1 variant with no ubiquitin present. The recorded spectra were referenced in TopSpin (version 3.6) and analyzed in CcpNmr Analysis (version 2.5.2) over the titration series. Chemical shift perturbations (CSPs) for the ^1^H-^15^N HSQC spectra were then exported, as calculated by [36]:(2)$$CSP=\sqrt{{\left(∆\delta_{H}\right)}^{2}+{\left(0.1\cdot \;∆\delta_{N}\right)}^{2}}$$
with ∆*δ_H_* representing the perturbation in the hydrogen dimension and ∆*δ_N_* the perturbation in the nitrogen dimension. CSPs for residues T39 and L40 of the Dss1 WT were used to fit and derive the maximum perturbation expected at the saturation of binding for the WT, ∆*δ**_max_*, as well as the dissociation constant, *K_D_*, based on the following relationship [36]:(3)$$CSP=∆\delta_{max}\frac{\left[P_{0}\right]\cdot \left[L_{0}\right]\cdot K_{D}-\sqrt{{\left(\left[P_{0}\right]\cdot \left[L_{0}\right]\cdot K_{D}\right)}^{2}-4\cdot \left[P_{0}\right]\cdot \left[L_{0}\right]}}{2\cdot \left[P_{0}\right]}$$
with [*P*_0_] and [*L*_0_] being the concentrations of Dss1 variant and ubiquitin, respectively. Then, using Equation (3) and in a global fit keeping the ∆*δ**_max_* fixed to the values found for the WT protein, the CSPs of T39 and L40 were fitted to derive the relative dissociation constant, *K*_D_, for each variant compared to WT Dss1. This relative *K*_D_ determination rather than absolute determination was performed as the ubiquitin titrations did not reach saturation, and intermediate exchange was observed in the Dss1 WT NMR experiments.

### 2.6. Small-Angle X-ray Scattering (SAXS)

SAXS data were collected at the DIAMOND beamline B21, London, UK, using a mono-chromatic (*λ* = 0.9524 Å) beam operating with a flux of 2 × 104 photons/s. The detector was an EigerX 4M (Dectris). The detector to sample distance was set to 3.7 m. Samples were placed in a Ø = 1.5 mm capillary at 288 K during data acquisition. The SAXS intensity profiles of the four proteins were measured at a temperature of 15 °C and a protein concentration of 3 mg/mL (WT, All-E) or 2 mg/mL (All-D, Swap). The average *R_g_* was calculated from the SAXS profiles using ATSAS 3.0.1 [37], using the Guinier approximation with a *q*_max_ corresponding to *q*_max_ ∗ *R_g_* = 0.9, which is commonly used for IDPs [38].

### 2.7. Diffusion-Ordered NMR Spectroscopy

Diffusion-ordered spectroscopy (DOSY) experiments [39] were performed on the four Dss1 variants (50 μM) to determine hydrodynamic radii (*R_H_*). The buffer was 20 mM Na_2_HPO_4_/NaH_2_PO_4_, 150 mM NaCl and 5 mM DTT, pH 7.4, 10% (*v*/*v*) D_2_O and 0.25 mM DSS. Translational diffusion constants were calculated by fitting the peak intensity decay within the methyl region (0.85–0.9 ppm), which was compared to the diffusion constant of the internal reference 1,4-dioxane (0.02% *v*/*v*) to estimate each protein’s *R_H_* as described [35]. Spectra were recorded (a total of 16 scans) on a Bruker Avance Neo 800 MHz spectrometer, with a Bruker carbon/nitrogen-optimized triple-resonance NMR observe cryoprobe with Z-field gradient over gradients strengths from 2 to 98% and using a diffusion time (Δ) of 200 ms and a gradient length of 3 ms (*δ*). Diffusion constants were fitted in GraphPad Prism v9.2.0.

### 2.8. Dss1 Peptide Assignment

The NMR resonances of Dss1 peptides (Dss1_51–69_) were assigned using spectra recorded via Bruker Topspin v3.6.2 on a Bruker 800 MHz spectrometer equipped with a cryogenic probe and Z-field gradient using natural isotope abundance (peptide concentration 1.2 mM, 20 mM Na_2_HPO_4_/NaH_2_PO_4_, 150 mM NaCl, 2% (*v*/*v*) trifluoroethanol, 10% (*v*/*v*) D_2_O and 0.25 mM DSS; pH 7.4), acquiring TOCSY, ROESY, ^15^N-HSQC and ^13^C-HSQC for manual assignments. Spectra were transformed and referenced using TopSpin v3.6.2 (Bruker, Switzerland) before being analyzed in CCPN Analysis v2.5 [32]. The fractional helicity of the peptides was calculated by averaging the Cα chemical shifts of residues D/E54 to G67, both included, and using 3.1 and 3.8 ppm as average reference chemical shifts for a fully formed helix (max *helix*), with the expression [40,41,42]:(4)$$\frac{\sum_{i}\left(∆\delta \;C_{i}^{\alpha }\right)}{\max helix}$$

### 2.9. Molecular Dynamics Simulations

Molecular dynamics simulations were performed with GROMACS v. 2019.6 [43,44,45,46] and plumed v. 2.6.1. [47,48,49] with the force field a99SB-*disp* [50]. Parallel bias metadynamics [51] with well-tempered bias potentials [52] was used for all simulations. The backbone dihedral angles and the *R_g_* of the C^α^ atoms were chosen as collective variables for biasing the simulations of full-length Dss1. The *R_g_* was biased within an interval of 1.3–4.0 nm. Only the backbone dihedral angles were biased for simulations of the Dss1 *C*-terminal-region peptides. We used multiple walkers [53] with 20 replicas per variant for the simulation of full-length Dss1. Gaussian “hills” were added to the bias potential at a frequency of 400 fs and a width determined using diffusion-based adaptive gaussians [54]. The bias factor was set to 32. A dodecahedral box was used, and periodic boundary conditions were applied. We used the a99SB-*disp* water model and added ions to a concentration of 150 mM. Simulations were run at a temperature of 283 K. Energy minimization was performed using a steepest descend algorithm followed by a conjugate gradient algorithm. The first equilibration was run for 2 ns with position restraints. The second equilibration was run for 2 ns without position restraints with pressure coupling using the Berendsen barostat. Constraints on bonds were applied with the LINCS algorithm [55]. For the equilibrations, constraints were applied to all bonds, and for the production, constraints were applied for bonds to hydrogen atoms. A cut-off of 0.9 nm was used for non-bonded interactions, and PME [56] was used for electrostatic interactions. A leap-frog integrator was used for the equilibration and production runs. The Parrinello-Rahman barostat was used in the production runs, and the velocity-rescale thermostat [57] was used for the equilibrations and production. Simulations of full-length Dss1 and variants were run for 13 μs (0.65 μs per replica) and the peptides for 8 μs with a timestep of 2 fs. We discarded the first 0.15 μs of each replica in the full-length Dss1 and 2 μs of the peptide simulations as equilibration, after calculating the accumulated bias for the entire trajectory, as the bias hereafter is considered static, allowing for reconstruction of the un-biased probability distribution using reweighting [58]. The reweighted and un-biased trajectories were used for the subsequent data analysis. Errors on the *R_g_* were estimated using block averaging [59,60]. The block size used for error estimation of the average *R_g_* of each simulation was the smallest block size in the plateaued region for the block error analysis of the free energy surface as a function of the *R_g_*.

Theoretical small-angle X-ray scattering (SAXS) profiles for each of the full-length protein simulations were calculated for every 50th frame (0.5 ns) with Pepsi-SAXS [61]. The theoretical profiles were compared to experimental SAXS profiles by reduced χ2 statistics. To account for uncertainty of the experimental errors, experimental errors were rescaled using a correction factor calculated with BayesApp v. 1.0 [62,63].

The secondary structure content of the MD-simulated conformations of full-length Dss1 and the helix region was calculated using the Dictionary of Secondary Structure of Proteins (DSSP) algorithm [64] using DSSP v. 2.2.1. DSSP assigns a secondary structure to a protein from the coordinates of the backbone atoms based on the possible hydrogen bonding patterns. Hydrogen bonding is defined by electrostatic interaction energy, and the cut-off is set as high, allowing the algorithm to pick up on hydrogen bonds that deviate from the ideal length and angle. Conformations classified by DSSP as α helix, 3_10_ helix or π helix were considered helical. DSSP was applied to all frames in the trajectories.

Contact maps for all simulations were calculated with the python package MDTraj [65]. Contacts between residues were defined with a distance cut-off of 8.5 Å between Cα atoms and calculated for every 10th frame in the trajectory. The contact maps were averaged and weighted by the metadynamics bias associated with each frame to arrive at a contact map representing the weighted fraction of simulation time that each residue pair was in contact. The differences in the contact maps between the wildtype and the variants were calculated as the log ratio of the fraction of contacts for each residue pair in the variant to those of the wildtype.

## 3. Results

### 3.1. The Alphabet of Disorder

We first examined how much the observed enrichment and depletion of the different types of amino acids in IDPs relative to a background of folded proteins depends on the chosen background dataset. Since, here, we define the enrichment of amino acids in the IDP sequence composition profile as the difference to the background normalized to the frequency of the amino acid in the background dataset, variations within the background dataset will affect the calculated enrichment. We examined three background datasets with different criteria for folded proteins: (1) the standard Composition Profiler folded protein dataset with high-quality X-ray structures, (2) a dataset with X-ray structures with low B-factors and thus, perhaps, less dynamic proteins, and (3) a broadly defined dataset including proteins with lower resolution (Appendix A). We found variations in amino acid frequencies in the three reference sets, and thus, the enrichment profile was dependent on the chosen background dataset, in particular for amino acids of low frequency. We decided that a broader definition of folded proteins would give us a more representative sequence profile for IDPs, because restricting the folded dataset to high-resolution globular proteins would also include the enrichment of amino acids in these structures (Appendix A). We chose a set of non-redundant and naturally occurring proteins that had an entry in the PDB as the best representation of folded proteins, which would include more diverse proteins than the standard Composition Profiler folded background dataset.

Next, we explored the composition profile in the DisProt database using the selected background data. Here, we found that the recent growth of DisProt and the use of the larger and more diversely defined folded protein dataset available in 2022 resulted in some differences compared to the earlier enrichment profile described by Uversky et al. 2013 [7] (Figure 2A). First, the main characteristics of the IDP sequence composition stands: a depletion in hydrophobic amino acids, an enrichment in polar and charged amino acids and an enrichment in the structure-disrupting amino acid proline. However, the most enriched and most depleted amino acids were now less extremely enriched or depleted. Glutamate thus appeared to be much less enriched compared to the original profile. A few amino acids shifted from being depleted or slightly enriched to being more enriched in IDPs, including asparagine, threonine, glycine, and aspartate. From a structural viewpoint, the enrichment of glycine in IDPs can be explained by the rotational freedom from the lack of a sidechain, allowing for a larger conformational space. Although not as pronounced as previously observed, we still observed a greater enrichment of glutamate and glutamine compared to the similar amino acids, aspartate and asparagine with a one-carbon-shorter sidechain.

Next, we investigated whether the observed differences could be explained by species-specific amino acid frequencies, as DisProt mainly contains eukaryotic proteins and sequences mostly from humans. To remove this potential bias, we created sequence profiles containing only eukaryotic or human sequences in both the folded and the disordered datasets. For all three sets, we observed a similar IDP composition profile (Figure 2B). However, we found a difference in the enrichment of glutamine in the species-specific profiles, indicating that the glutamine enrichment in IDPs might partly be explained by a depletion of glutamine in prokaryotes compared to eukaryotes. There was no substantial difference in the enrichment of glutamate and aspartate in the species-specific composition profiles, and we could thus not explain the difference in glutamate and aspartate enrichment by a species bias.

### 3.2. Functional Effect of Aspartate and Glutamate in Dss1

To investigate whether this apparent bias towards glutamate in IDPs would relate to functional effects, we used the small acidic IDP from *S. pombe,* Dss1, which is a component of several different protein complexes [19,22], including the 26S proteasome [66,67,68], and which can bind both mono- and poly-ubiquitin. Dss1 is overall highly negatively charged (−18) with a distributed content of both glutamates (9) and aspartates (14), totaling 23 negative charges. We designed three variants with different Glu/Asp ratios: an All-E variant, where all 23 acidic residues were glutamates, an All-D variant where all were aspartates and a Swap variant where we exchanged glutamate for aspartate and *vice versa* (Swap). Together with the wildtype (WT) protein, we first assessed the functional effect of the aspartate-glutamate substitutions.

#### 3.2.1. The Glu/Asp Variants Are Functional in Vivo

To investigate whether the Glu/Asp variants of Dss1 retained function, we tested the ability of overexpressed GFP-tagged versions of the Dss1 variants to rescue the temperature-sensitive growth defect of a Dss1 knockout strain (*dss1*Δ). First, we tested the expression of the recombinant Dss1 variants by analyzing whole-cell extracts via SDS-PAGE and Western blotting. This revealed that all variants were expressed at roughly equal levels (Figure 3A). As shown before [23], the *dss1* null mutant is viable at 29 °C, but unable to form colonies at 37 °C (Figure 3B). In all cases, we observed that the overexpression of the recombinant *dss1* variants suppressed this temperature-sensitive growth defect as efficiently as WT (Figure 3B). We conclude, therefore, that any changes in the conformational ensembles of the variants are too subtle to substantially impair the Dss1 function relevant to this phenotype in vivo. We note, however, that this phenotype complementation assay may not be sensitive enough to capture small effects and that the temperature-sensitive phenotype of the *dss1*Δ strain is primarily linked to a lack of Dss1 incorporation in the 26S proteasome [23,66,67,68]. The assay therefore does not report on the other cellular functions of Dss1.

#### 3.2.2. Ubiquitin Binding Affinity, but Not Binding Ability, Depends on Glutamate

We then used NMR spectroscopy to assess if all Dss1 variants could bind to ubiquitin. This was carried out by quantifying changes in the chemical shifts (chemical shift perturbation, CSP analysis) in a ^15^N-HSQC NMR spectrum after the addition of 40 molar excess of mono-ubiquitin (Figure 4); the chemical shifts of each variant were first assigned using sets of triple-resonance 3D NMR spectra. Overall, the same residues in the variants were affected by the addition of ubiquitin, confirming the binding of ubiquitin to all four variants (Figure 4A). However, binding to UBS I in Dss1 led to the disappearance of peaks in the spectra, mainly of the WT, but also to a much lesser degree in the All-E variant, indicating exchange between free and bound states on an intermediate NMR time scale.

The broadening or loss of signals makes it difficult to quantify binding affinity; thus, in a recent study, we titrated ^15^N-ubiquitin with unlabeled Dss1 to circumvent the loss of signal intensity to quantify the affinity of WT Dss1 for mono-ubiquitin giving a *K_d_* of 380 μM [24]. The titration of the variants with ubiquitin into 40 molar excess showed smaller CSPs than WT Dss1 (Figure 4B and Appendix A), suggesting weaker affinities and a smaller population of the bound state. However, since ubiquitin is known to form dimers at mM concentrations [69], we were unable to reach saturation. Instead, here, we determined the fold-change in affinity from global fitting to the CSPs of all variants using data from the same residue, either T39 or L40 (Figure 4B). All variants bound mono-ubiquitin 3.5–7.5-fold weaker than WT Dss1, with the Swap variant having the lowest affinity. The sequence of UBS I in WT contains a central glutamate and is flanked by two aspartates on each side. The WT and All-E variant both have this central glutamate in common, while the two other variants have an aspartate in this position. Thus, the overall weaker affinity suggests a combined preference for glutamate in one specific position within the UBS I motif with flanking aspartates for increased affinity. However, the weaker affinity of the variants for mono-ubiquitin may be rescued to some extent when binding to ubiquitin chains, due to avidity and a local concentration effect, thus possibly explaining the lack of a phenotype in yeast.

### 3.3. Global Compaction Does Depend on Glu vs. Asp Ratio

To assess chain compaction of the four variants, we determined the hydrodynamic radius, *R_H_*, of the four Glu/Asp Dss1 variants via pulse field gradient NMR and the radius of gyration, *R_g_*, via SAXS. In parallel, we performed all-atom molecular dynamics (MD) simulations of each of the four Dss1 variants, applying enhanced sampling techniques to push the simulations to explore more conformations in the ensemble (Figure 5). We used parallel bias metadynamics [51] and chose the *R_g_* and the dihedral angles as collective variables to increase the sampling of backbone conformation without directly biasing the simulations towards a helical conformation. We calculated the average *R_g_* from the MD simulations of each variant by calculating the *R_g_* from the coordinates of the atoms in each frame. We then compared the dimensions of the four variants (Figure 5B).

First, we found that the global dimensions of Dss1 as extracted via NMR diffusion and SAXS were overall relatively similar for all Glu/Asp variants of Dss1, although we note that the *R_H_* for the All-E and Swap variants were slightly larger than that of WT Dss1 (by 17% and 13%, respectively). Second, we compared the *R_g_* calculated from the MD ensembles to the *R_g_* measured via SAXS and found no substantial differences between the variants nor between the simulation or experiment, except for the simulation of the Swap variant, which showed a slightly more expanded chain by MD. Additionally, we calculated the theoretical SAXS intensity profiles from the MD ensembles using Pepsi-SAXS and compared the profiles directly to the experimental SAXS intensity profiles, which showed similar agreement between the simulation and experiment (Appendix A).

From the simulations, we also calculated the average number of contacts between C^α^ residues within the protein (Figure 5C) and were able to observe contacts between the C-terminal region and the UBS I, in agreement with conclusions from a study using paramagnetic relaxation enhancement NMR [19]. In our MD simulations, we observed that this interaction primarily took place between the last part of the region that also samples helical structures, where there are three consecutive lysines, and the UBS I. We also observed that interactions between the C-terminal region and the UBS I were more frequent in the WT and All-D, but nearly absent in the All-E variant. This implies that the aspartates on both sides of the binding site facilitate this interaction, which could, as proposed in [19], be a mechanism to regulate the accessibility of the binding site. While the effects are small, we also note that both the *R_H_* and *R_g_* values suggest that the WT is the most compact variant.

### 3.4. Local Structural Changes in Dss1 Depending on Glu/Asp Variants

Since the global properties of Dss1 appeared to be mostly indifferent to either glutamate or aspartate, but with changes in contacts between the C-terminal and the UBS I, we asked if the glutamate bias would be explained by effects on local structure formation in Dss1. To answer this question, we analyzed the secondary chemical shifts (SCS) of the C^α^ and C^β^ atoms in all four Dss1 variants (Figure 6). Additionally, we analyzed the local structure in the MD ensembles by calculating the most likely hydrogen-bond patterning based on the distances between atoms in each frame with DSSP and simulated the C-terminal region from residue 50 to 71 of the four proteins alone, allowing us to run longer simulations and observe the formation and unfolding of the helix in a single trajectory.

All four Dss1 variants showed the formation of a transient C-terminal α helix, which was evident both from NMR SCSs and from the MD simulations of both full-length Dss1 and the C-terminal helical region (Figure 6). From the NMR secondary chemical shifts, we found that the largest difference between the Glu/Asp variants and WT Dss1 was in the population of the transient C-terminal α helix, where all variants have a smaller α helix population compared to wildtype variants, as well as in a region capping the N-terminal side of the helix (Figure 6). Here, WT and All-D had a short stretch of negative C^α^ SCSs indicating an extended structure, which was absent in the Swap and All-E variants. The extended structures of the disordered ubiquitin binding sites were maintained, but it appeared that consecutive aspartates increased the extendedness compared to consecutive glutamates. Thus, locally, there appears to be an effect of Glu/Asp variation where aspartate may better support the local structure of extended characters.

While the MD simulations do capture the formation and unfolding of the C-terminal transiently populated α helix in all variants, we did not observe a larger population in WT Dss1. In the simulations of the C-terminal regions, we observed that the formation and unfolding of the α helix is a slow process even when applying a bias against already visited conformations, where the process takes around 1 μs (Appendix A); we note here that the bias that is used to enhance sampling means that the kinetics and mechanism of helix formation/breaking is perturbed. Possibly, the simulations of full-length Dss1 of 10 μs do not therefore capture the population of the C-terminal α helix precisely. However, the populations of the C-terminal α helix are similar in both the simulations of the C-terminal peptides and the full-length Dss1 variants, indicating that the true population can be expected to be in between these populations.

In the MD simulations, UBS I appears to be transiently helical (Appendix A). As this is not observed in the NMR SCS analysis, it is perhaps a result of remaining force field inaccuracies, for example, related to the solubility of hydrophobic amino acids [50]. When the helix is formed, two tryptophans are positioned across from each other, and their hydrophobic interaction is slightly stronger, thus perhaps overstabilizing the helical conformation. In the C-terminal transiently helical region, there are no tryptophans, and thus, we did not expect this issue to have a major impact on the helix population in this region.

In the simulations, we observed that the full eleven-residue C-terminal helix rarely forms, while either one or the other half of the helix forms more frequently (Appendix A). While this could be because of the slow formation of the full helix, it could also describe the same transient helicity as the NMR secondary chemical shifts, as these represent a bulk average and could thus have contributions from conformations with either end of the helix formed. We did not observe a direct effect on helix population of the helix-capping residues being either glutamate or aspartate in the MD simulations but observed that glutamates (residue 52–54) positioned before the helix in the All-E and Swap variants are more frequently helical at these positions than the aspartates in the wildtype and All-D variants. This might indicate that glutamate supports a helix conformation better than aspartate, or that aspartate is more frequently found in a helix-capping conformation and thus does not have a helical geometry. We also note that in the simulations, in some cases, we observed a small dip in the average fraction of helical structure near the middle of the helix; this did not appear to be observed in the experiments. A more direct comparison, however, would require better methods for calculating small changes in secondary chemical shifts from simulations.

The most pronounced effect of Glu/Asp variations on the local structure was observed for the helix population, which changed by just minor alterations to the amino acid composition. We therefore decided to examine specific single-amino-acid substitutions that we hypothesized would either increase or decrease the helix population. To be able to capture these minor changes in population in an otherwise disordered chain, we analyzed the effects using peptides corresponding to the helical 19-residue C-terminal region of Dss1, _51_GDDDFSVQLQAELKKKGVA_69_. From the MD simulations of the full-length Dss1 proteins, we observed that the lysines K65 and K66 often formed salt bridges with E62 in the WT and All-E variants more frequently than the aspartate E62D in the Swap and All-D variants, which formed salt bridges with K64 more often (Appendix A). When the residues are in an α-helical conformation, the sidechain of D62 will be in proximity of the residues K65 and K66, while K64 would be on the other side of the helix. Salt bridges between E62 and K65 and K66 thus likely form helix-stabilizing interactions, while conformations with salt bridges between E62 and K64 are unlikely to be α-helical. We thus speculated that the interaction between K65 and E62 could stabilize the helix, and that the sidechain of aspartate is likely too short to support this interaction. The NMR chemical shifts indicate the possible helix-capping function of D52–D54, which could also stabilize the helix. This would explain why the population of the helix is largest in the WT, as it both has a glutamate at position 62 and aspartates at positions 52–54. Based on these observations, we designed the following peptides (residue 51–69): WT (12% helicity predicted by Agadir) and two helix-modulating D/E-Swap variants, D54E (26% predicted helicity) and E62D (9% predicted helicity). Using NMR spectroscopy, we extracted helicity from the SCS for the C^α^ atoms (Figure 7, see Materials and Methods). Here, we found that although not reaching the full effect, the predicted effects of the substitutions were captured experimentally, with the D54E variant increasing in helicity (from 27.7% to 29.3%), mostly at the substitution site, and the E62D losing helicity (from 27.7% to 20%). Thus, in the peptides, the D54E gained 6% in helicity and the E62D lost 28%. This suggests that the substitution of aspartate for glutamate in the *N*-terminal of the helix increases helicity and removing a glutamate in the middle of the helix destabilizes it. Thus, these data support that glutamate is preferred for the stabilization of this transient helix.

## 4. Discussion

The enrichment of glutamate over aspartate in IDPs has been known since the early 2000s, but to our knowledge, no systematic attempt to explain this bias has been performed. Since then, current databases on IDPs have expanded, and additional proteomes have become available, enabling us to revisit the basis for this bias. Using a disorder dataset containing five times as many sequences and comparing to a more diverse dataset of folded proteins, we found that while glutamate is indeed enriched in IDPs, the difference is less pronounced than originally found. We also exclude the possibility that this difference is due to an underlying bias caused by the dominance of human proteins in the disorder database.

Using yeast Dss1 as a model IDP, we sought a functional explanation for the enrichment of glutamate over aspartate. We found, however, that all tested Dss1 variants were able to complement the growth defect of a *dss1* deletion mutant, irrespective of the type of anionic sidechain. Since previous studies have shown that the temperature-sensitive phenotype of the *dss1*Δ strain is tightly connected with the incorporation of Dss1 into the 26S proteasome [23,66,67,68], it is likely that the structural and dynamic effects of the Asp/Glu substitutions are not sufficiently pronounced enough to disrupt its interaction with the 26S proteasome in vivo. However, we note that the Dss1 variants were GFP-tagged and overexpressed, which could mask subtle effects. In addition, Dss1 is also involved in other cellular processes and is often found as a subunit in larger complexes [19]. Since the effects of these functions of Dss1 and more subtle effects on proteasome incorporation may not have been captured by our growth assays, we cannot rule out the concept that some Dss1 functions are not affected by Asp/Glu substitutions.

As also suggested by the growth assays, we observed that all variants were capable of binding mono-ubiquitin. Although exchanging the anionic amino acids did not impair the ability of Dss1 to bind to ubiquitin, variants bound mono-ubiquitin 3.5–7.5-fold weaker. Since Dss1 prefers ubiquitin chains over mono-ubiquitin, avidity in binding may rescue some of this effect, and hence may not therefore lead to any phenotype. Additionally, stronger binding to ubiquitin appears to be a combined result of expanding the binding region from the flanking aspartates and optimizing affinity in UBS I by the glutamate. In a study on Dss1, Schenstrøm et al. found that the C-terminal region of Dss1 bends back and shields the UBS I and suggested that this may be a mechanism for regulating ubiquitin binding [19]. In our MD simulations, we were able to observe this interaction for all but the All-E variant. Interestingly, we found that the more aspartates the variants contain in the UBS I, the more frequent are the interactions between the C-terminal region and the UBS I. Similarly, in a recent study, Zeng et al. found that aspartate in IDPs forms hydrogen bonds more frequently than glutamate, likely stabilizing observed local chain compaction [70].

No large differences in *R_H_* or average *R_g_* in the simulated conformational ensemble and SAXS experiment were observed, and glutamate was therefore not found to promote substantially more expanded ensembles than aspartate. Only minor differences in local structure were observed via NMR, where consecutive aspartates better support extended structures than consecutive glutamates, and the positional effects of having a Glu or an Asp can influence the population of transient helices. Although these small population changes were detectable via NMR, the effects were likely too small to be manifested and detectable in the *R_H_* or *R_g_* measurements. Likely, for Dss1, which is already highly charged, changing the type of anionic sidechain will have little effect on the net charge per residue and hence may not affect chain collapse [71].

Instead, we observed that local structural effects and binding strength could be modulated by exchanging Glu/Asp preferences. For Dss1, the transient helix in the C-terminal region was populated differently in the Glu/Asp variants. Using peptide variants, we could establish that glutamate stabilized the transient helix both when positioned near the N-terminus and when inserted into the helix at positions that enable salt bridge formation with positively charged sidechains positioned three residues C-terminally of it. This is consistent with previous work on folded proteins, where glutamate is more frequently observed in α helices than aspartate [9] and where glutamate in the center of a helix is generally more stabilizing than aspartate, because the carboxyl group is more distant to the backbone. Thus, glutamate imposes less restraints on the conformational space of the residues in the helix [10].

Our work has explored a potential link between a Glu/Asp bias in IDPs, local conformational preferences and functional effects. A question remains as to when and why evolution would favor glutamate over aspartate in IDPs. Recent work has shown that higher helicity in the free state of an IDP may lead to higher affinity for a partner protein [12,72]. Combined with the preference for glutamate seen here to stabilize transient helices, this could suggest that glutamate would be a preferred residue for highly populated transient helices in IDPs. The preformation of highly populated helices could be important in folding-upon-binding reactions, where increased helicity has been shown to affect affinity through effects on both *on*- and *off*-rate constants [12,72]. However, aspartate is a known helix N-capping residue [73], and has been shown in several IDPs to initiate helices, even at several positions within the same helical stretch [74,75], but a quantitative comparison of the two amino acids for this property has to our knowledge not been performed. Finally, in a recent work studying the interactions between IDPs and calcium ions, a preference for aspartate over glutamate in the so-called Escaliber-like motif was noted [29]. Thus, another possible reason for suppressing the use of aspartate in IDPs would be to minimize the binding of divalent cations.

While our results point towards a bias arising from the function of the anionic amino acids in IDPs, the difference in the enrichment of glutamate and aspartate could also arise through other multiple local effects, explanations to which need exploring. The subtle differences in the choice of amino acid at specific positions may be able to shift the equilibrium populations of the conformational ensemble in IDPs and thus impact their function and interactome.

## 5. Conclusions

Here, we addressed the compositional bias in IDPs, which have preference for glutamates over aspartates, a phenomenon pointed out already in the early 2000s [4,6,7,8]. We found the dimension of the disordered Dss1 is largely indifferent to the difference between these anionic amino acids, whereas highly local effects both on the populations of transient structures as well as on binding affinity were seen. We hypothesize that stabilizing local transient helix structures through capping effects and intra-helix salt bridges, as well as adding binding strength through the additional methylene group, may be important reasons for the preference of glutamates over aspartates in IDPs. Finally, functional biases towards glutamates in regions undergoing helical folding-and-binding and towards aspartates in transactivation domains and calcium-binding regions are likely just some of several functional reasons for the selection of glutamate or aspartate in specific IDPs.

## Figures and Tables

**Figure 1 biomolecules-12-01426-f001:**
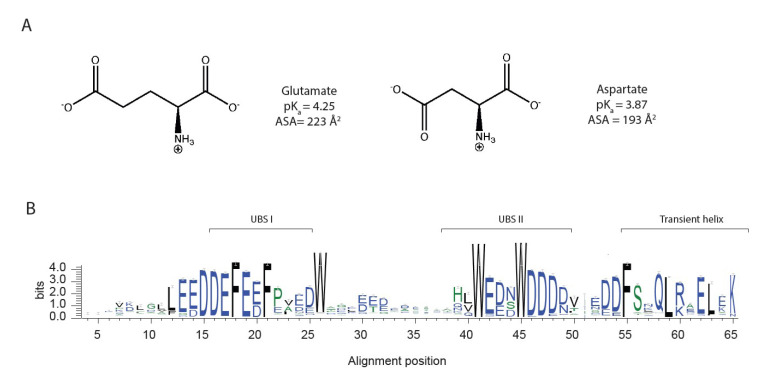
Glutamate and aspartate have similar chemical properties. (**A**) Chemical structure of the amino acids, glutamate and aspartate, (**B**) graphical representation of multiple sequence alignment of the Pfam [17] Dss1/Sem1 family (PF05160), only showing the positions that are present in *S. Pombe* Dss1, with information content on the y-axis (for the full MSA, see Appendix A). The sequence logo was made using WebLogo 3 [18].

**Figure 2 biomolecules-12-01426-f002:**
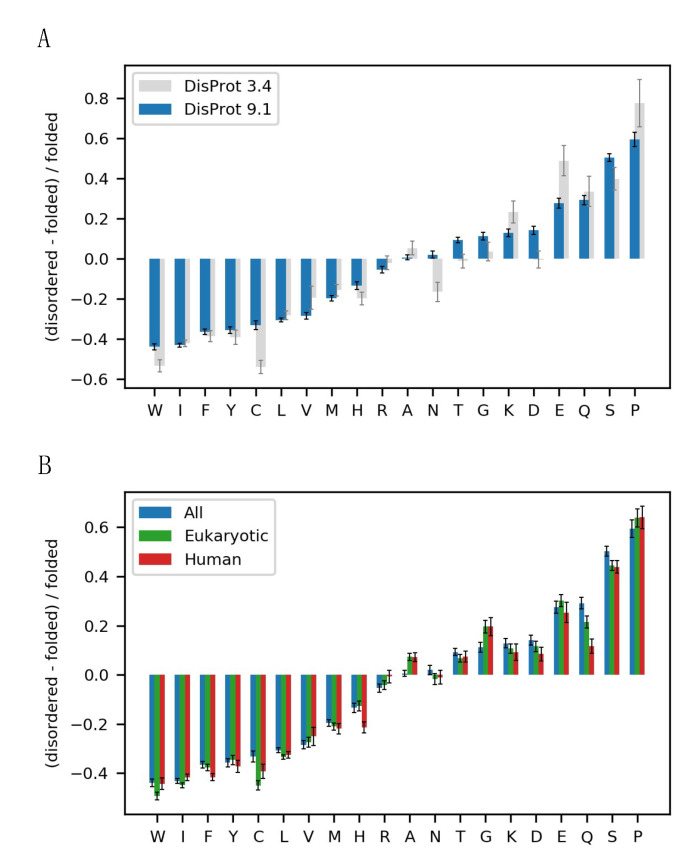
The disorder alphabet revisited. (**A**) Sequence enrichment profile of IDPs with the enrichment of amino acids in disordered proteins from the newest (9.1) and the older version (3.4) of DisProt [13,14,15] compared to folded proteins, defined, respectively, as non-redundant sequences in the PDB or the standard Composition Profiler dataset PDBselect25. (**B**) Sequence enrichment profiles for eukaryotic and human IDPs from DisProt v. 9.1 using a background set of non-redundant sequences in the PDB. Error bars show the boot strap confidence intervals with a statistical significance value of 0.05.

**Figure 3 biomolecules-12-01426-f003:**
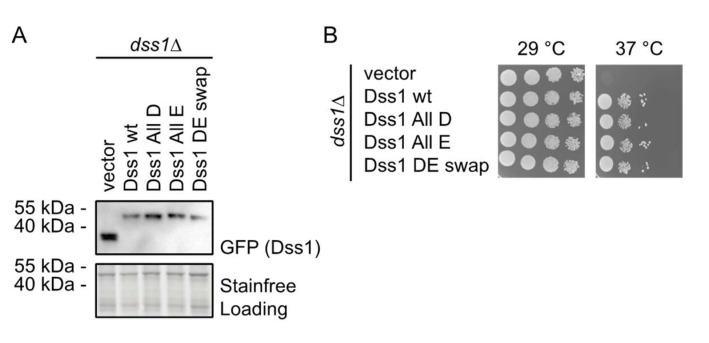
Growth effects of Asp and Glu substitutions in Dss1. (**A**) *S. pombe* cells deleted for Dss1 (*dss1*Δ) were transformed to express GFP-tagged wildtype (wt) Dss1 and the indicated Dss1 variants. Whole-cell lysates were then compared via SDS-PAGE and Western blotting using antibodies to GFP. Stain-free labeling was used as a loading control. (**B**) The growth of the strains from (**A**) was compared by serial dilution and incubation on solid media at 29 °C and 37 °C. Note that the temperature-sensitive growth defect of the *dss1*Δ strain is rescued by all Dss1 variants.

**Figure 4 biomolecules-12-01426-f004:**
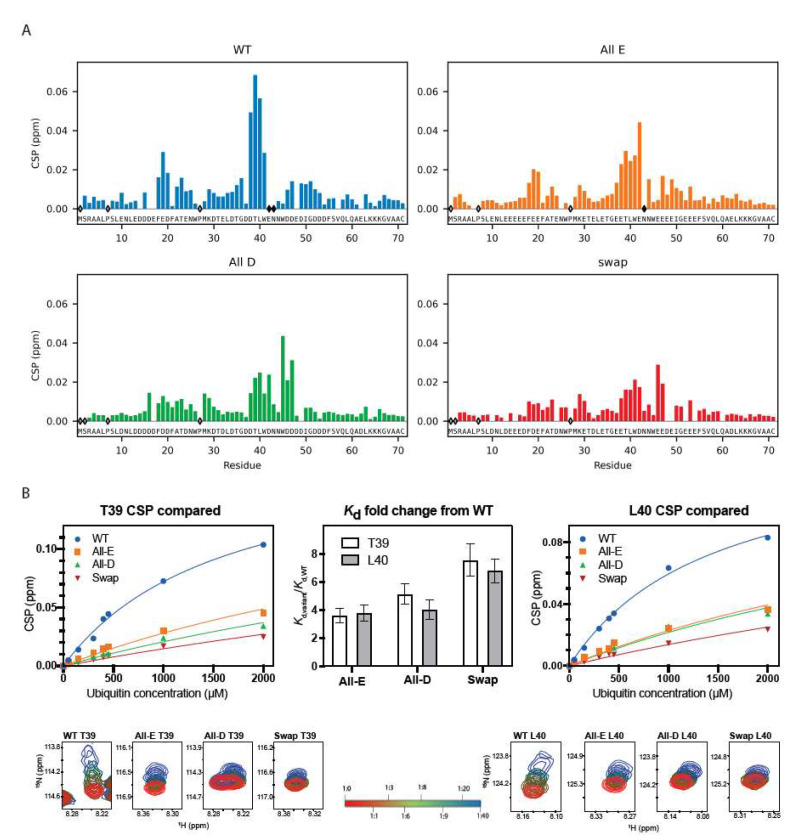
Ubiquitin binding ability is supported by both glutamate and aspartate. (**A**) CSP for Dss1 WT and Glu/Asp variants (each at 50 µM concentration) binding to a 40 times molar excess of ubiquitin (2 mM). Empty diamonds indicate non-assigned residues, and filled diamonds indicate disappearance of peaks upon ubiquitin addition. (**B**) CSP of Dss1 WT and Glu/Asp variants for residue T39 (left) and L40 (right) as a function of ubiquitin concentration, fitted to derive the relative affinities for each variant. The middle bar plot shows the fold-change in affinity for the three variants (compared to WT Dss1) derived from the fits to either T39 or L40. The bottom figures show the HSQC peaks for T39 (left) and L40 (right) during ubiquitin titration (from 1:1 to 1:40) indicated by the red-green-blue color change, respectively.

**Figure 5 biomolecules-12-01426-f005:**
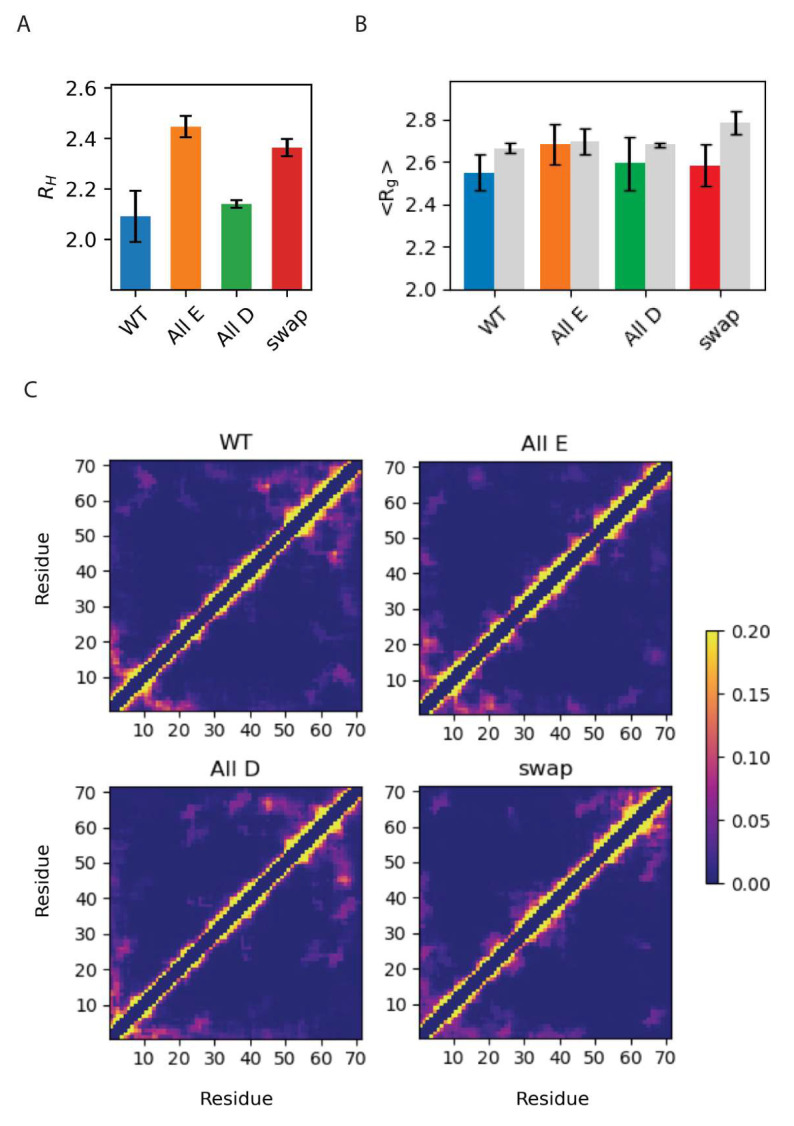
Global chain properties of Dss1 WT, All-E, All-D and Swap. (**A**) The hydrodynamic radius (nm) determined via diffusion NMR; (**B**) average radius of gyration (nm) determined experimentally via SAXS (color) or via MD simulations (gray); (**C**) contact maps showing the frequency of contacts between each of the 71 residues in the four Dss1 variants in the MD simulations, where yellow corresponds to an interaction found > 20%.

**Figure 6 biomolecules-12-01426-f006:**
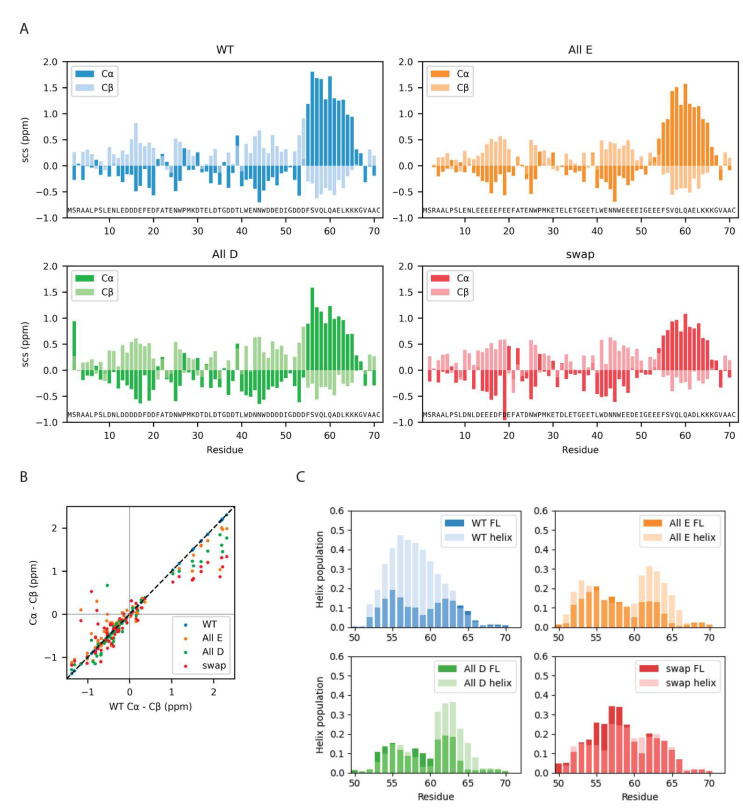
Local chain properties of Dss1 WT, All-E, All-D and Swap. (**A**) NMR secondary chemical shifts (C^α^ and C^β^) for Dss1 wildtype and the three variants; (**B**) correlation between SCSs of WT Dss1 and the three variants; (**C**) helix population of the C-terminal region from MD simulations of full-length (FL) Dss1 overlayed with the helix population from the simulations of the helix region alone.

**Figure 7 biomolecules-12-01426-f007:**
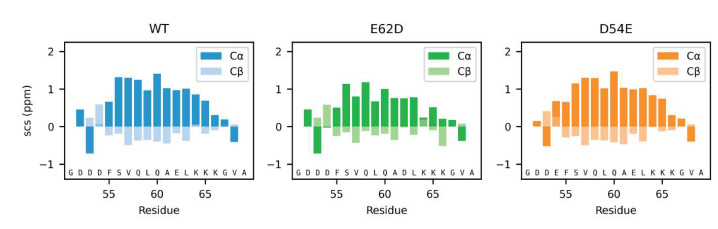
Local effects of Asp and Glu in helix stabilization. The SCS of the Cα and Cβ atoms of the three peptides corresponding to WT, D54E and E62D in the context of Dss1_51–69_.

## Data Availability

MD simulations and SAXS data will be available via https://github.com/KULL-Centre/_2022_Roesgaard-Lundsgaard_DSS1, accessed on 16 August 2022. Chemical shifts for all four Dss1 variants are available at BMRB, IDs:27618, 51551, 51552 and 51557.

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
