# Peer review of "Deciphering the Alphabet of Disorder—Glu and Asp Act Differently on Local but Not Global Properties"

_biomolecules, 2022, doi:10.3390/biom12101426_

Round 1

Reviewer 1 Report

In the manuscript by Kragelund and colleagues entitled “Deciphering the Alphabet of Disorder - Glu and Asp Act Differently on Local but Not Global Properties” the authors investigate the structural and functional effects of Glu and Asp substitutions in IDRs using Dss1 as a model. Dss1 binds to the proteosome and ubiquitin and other partners? It is conserved in S. Pombe where the authors overexpressed WT and 3 variants of Dss1 with engineered Glu/Asp swaps and saw no appreciable effect on phenotype in a temperature sensitive growth assay that depends on Dss1 binding to the proteosome.

Based on the conservation of Dss1 for the three motifs shown in Figure 1 I expected at least a detectable phenotype. If the sequence doesn’t matter why is it conserved? Do you have a marker or assay for proteosome function? Does Dss1 have a secret paralogue?

I was unable to find any Kd values based on equation 3. Does this mean your assessment of affinity is based on the differences in the shifts you see for the variants after adding a 40-fold excess of ubiquitin? Based on the chemical shift differences and the concentration of ubiquitin do you think any of these differences is greater than 0.1 kcal/mol? What happens in vivo? Is there a PTM that makes Dss1 bind ubiquitin more tightly or does the transient helix bind the proteosome? It seems like some basic details are missing about the system.

Referring to Figure 6c: is it normal for MD to capture helix formation at the ends instead of in the  middle? I have never see a SCS profile for a transient IDP helix with a dip in the middle unless two helices are forming but it seems like one helix forms in the bound state. What does this helical region bind to, the proteosome?

Why are the SCS values different than 6A? For instance, D52 and D53 for WT both have negative CA SCS in 6A but D52 is positive in Figure 7. Did you recalculate the random coil shifts based on a shorter sequence?

I think the conclusions on lines 525-527 may be a bit strong. If the effect is so pronounced you would have expected to see it in the growth assays. Can you see transcriptional activation in a two-hybrid assay between Dss1 and ubiquitin?

I don’t agree with the conclusion on lines 552-556. Do you really think the helix is forming from both ends (not the middle) after the transition state forms?

Why is there so much empty space on pages 10 and 12?

Figure 1 needs to be adjusted so all the letters are readable. W does not look like a W.

In the abstract: “interactions using model acidic IDP” should be “interactions using a model acidic IDP”

Some additional editing for spelling and grammar is warranted.

Reviewer 2 Report

Ahrensback Roesgaard et al. present a manuscript in which they address the apparent bias of glutamate (E) and aspartate (D) enrichment in intrinsically disordered proteins (IDPs). The authors first engage in a a bio-informatics analysis, by which they investigate the impact of the pool of folded proteins to which the sequence compositions of IDPs are compared. The authors show that the difference in D and E enrichment clearly varies with the choice of the pool of folded proteins. The pool the authors choose as a ‘best representation of folded proteins’ combined with sequence data of the most up to date DisProt database leads to the conclusion that E is much less favored over D in IDPs than previously anticipated. E remains, however, slightly more enriched in IDPs than D.

The authors then engage in various experimental and computational techniques to investigate potential functional differences that D and E may have in IDPs. The authors resort to the protein Dss1, which they have previously characterized, and its variants that have all D replaced by E (all E), all E replaced by D (all D), or that have D and E swapped (DE swap). The protein is entirely disordered, comprising two ubiquitin binding sites (UBS) and a C-terminal transient helix that are all highly enriched in both D and E (interestingly the content of D is higher in this wild type protein!).

While no changes are detected in yeast growth assays, by which a dss1 knockout strain is complemented with either the wt or the mutant Dss1 variants, a structural analysis of Dss1 using nuclear magnetic resonance (NMR) spectroscopy, small angle X-ray scattering (SAXS) as well as molecular simulations demonstrates only minor differences between the protein variants. The authors attribute these differences to a preferred use of E for helix stabilization and of D for helix capping, which also affects binding affinities.

Even though this explanation seems reasonable in view of the experiments undertaken, these hypotheses need to be strengthened by further experiments/analyses and the points below addressed.

Major points:

  • Based on NMR experiments, the authors state that the affinity of Dss1 binding to ubiquitin is altered in the different Dss1 variants compared to the wild type protein. The authors observe a signature of intermediate exchange in the titration of Dss1 wt and the all E variant and use this as an argument for a higher affinity of these two proteins compared to the other variants. The statement on affinities of the different Dss1 variants towards ubiquitin needs to be strengthened. The authors should show titration data for the variants (as shown for the wild type in Figure 4A) and a quantitative analysis of affinities should be undertaken. Interestingly, the methods section contains an explanation of how affinity constants are fitted from chemical shift titrations, but those titrations or their fits are not shown anywhere (the ‘most perturbed residues in the Dss1 variant’ should be named explicitly). Neither are the affinity constants mentioned. The authors should include those into the manuscript. An analysis of the affinity constant for binding of Dss1 wild type and all E variant should also be attempted even if in intermediate exchange, for example employing NMR exchange techniques. If it is easier, other methods to retrieve affinity constants could also be used, for example isothermal titration calorimetry (ITC). A quantitative analysis of affinities is indispensable to support the authors’ claim on affinity differences.

  • The effect on D and E for secondary structure and in particular population of the C-terminal transient helix is a central point in the manuscript and is used to support the hypothesis that E is preferentially used for helix stabilization and D for helix capping. Even though the NMR SCS show only small differences between the different variants, a conclusion in the direction proposed by the authors seems reasonable. However, the statements of helix population from the simulation data seems more questionable as also the UBS I is supposed to populate significant helical conformations (Figure S7), which, according to the experimental data, is clearly not the case. How much can the authors thus rely on the prediction of secondary structure in the C-terminal helix region? Also the argument that E positioned before the helix (all E and swap variant) leads to higher helical propensity does not seem to be supported by the experimental data, where the swap mutant shows overall less helical propensity. The authors should take care to carefully interpret and compare the NMR and simulation data and also clearly state where those are not in agreement. The statement ‘the NMR chemical shifts and the secondary structure calculated by MD simulation indicate possible helix capping function of D52-D54’ seems also rather poorly supported from the MD side.

  • The D54E mutant and the E62D mutant, inspired from the MD simulation data, show, as indicated by the authors, only very subtle differences in SCS compared to the wild type protein. The authors state that ‘the D54E gained 6%-point in helicity and the E62D lost 28%-point’. Does this refer to the overall helicity in the wild type? The authors should be clearer on that and also explain how overall hecility was calculated.

  • Overall, even if the effects are subtle, the NMR data seem to support the hypothesis that E is used for helix stabilization and of D for helix capping in Dss1. However, those subtle effects stem from the analysis of only a single helix in one protein. Would an analysis of known transient helices in IDPs for their D/E content and their positions within the helices help to support the hypothesis (particularly as D is a known N-capping residue as also the authors state)?

Minor points:

  • The authors state: ‘One reason for the enrichment of glutamate could be that glutamate has disorder promoting properties distinct from those of aspartate.’ Is this a hypothesis? In this case it should be phrased as such, otherwise the authors need to explain why that would be the case and insert citations.

  • The authors should provide spelled out names for all the abbreviations used in the manuscript (e.g. chemicals in the methods section).

  • Numbering of the supporting Figures is wrong throughout the supporting information, this needs to be corrected.

  • Figure S2 seems to be missing one panel: ‘a broadly defined dataset including proteins with low resolution’. Dividing this figure into (A), (B) and (C) and corresponding referencing in the main text may also help to understand the different data sets.

  • Figure 2A: Is it right that DisProt 9.1 is compared to the non-redundant sequences in the PDB and DisProt 3.4 to the Composition profiler dataset PDBselect25? It would be interesting to see the two versions of DisProt compared to the same dataset of folded proteins

  • The authors observe a difference in the hydrodynamic radii of Dss1 variants compared to wild type, but not in the radii of gyration. Can the authors explain this?

  • Line 431: Do the authors mean ‘negative C-alpha SCSs’?

  • Concerning their simulations, the authors state that ‘Further, the full eleven residue helix rarely forms, while either one or the other half of the helix forms more frequently (Figure S7).’ This is not what Figure S7 seems to show as no information of simulation time is contained in Figure S7.

  • Line 563: ‘IDPs’ rather than ‘DPs’?
